# Synthesis of encapsulated ZnO nanowires provide low impedance alternatives for microelectrodes

**Mohsen Maddah**[1,2]*, **Charles P. Unsworth**[2,3º], **Gideon J. Gouws**[2,4], **Natalie O. V. Plank**[1,2º]

**1** School of Chemical and Physical Science, Victoria University of Wellington, Wellington, New Zealand, **2** The MacDiarmid Institute for Advanced Materials and Nanotechnology, Wellington, New Zealand, **3** Department of Engineering Science, University of Auckland, Auckland, New Zealand, **4** School of Engineering and Computer Science, Victoria University of Wellington, Wellington, New Zealand

º These authors contributed equally to this work.
* mohsen.maddah@outlook.com

**Data Availability Statement:** All relevant data are within the article and its Supporting Information files.

**Funding:** Mohsen Maddah: This research was supported by the Royal Society of New Zealand Marsden Fund (3709273/UOA1510).

## Abstract

Microelectrodes are commonly used in electrochemical analysis and biological sensing applications owing to their miniaturised dimensions. It is often desirable to improve the performance of microelectrodes by reducing their electrochemical impedance for increasing the signal-to-noise of the recorded signals. One successful route is to incorporate nanomaterials directly onto microelectrodes; however, it is essential that these fabrication routes are simple and repeatable. In this article, we demonstrate how to synthesise metal encapsulated ZnO nanowires (Cr/Au-ZnO NWs, Ti-ZnO NWs and Pt-ZnO NWs) to reduce the impedance of the microelectrodes. Electrochemical impedance modelling and characterisation of Cr/Au-ZnO NWs, Ti-ZnO NWs and Pt-ZnO NWs are carried out in conjunction with controls of planar Cr/Au and pristine ZnO NWs. It was found that the ZnO NW microelectrodes that were encapsulated with a 10 nm thin layer of Ti or Pt demonstrated the lowest electrochemical impedance of 400 ± 25 kΩ at 1 kHz. The Ti and Pt encapsulated ZnO NWs have the potential to offer an alternative microelectrode modality that could be attractive to electrochemical and biological sensing applications.

## 1. Introduction

Microelectrodes, also commonly known as ultramicroelectrodes, are electrodes with dimensions in the micrometer range. They benefit from their decreased ohmic drop of potential, the fast establishment of a steady-state signal, a current increase due to enhanced mass transport at the electrode boundary, increased signal-to-noise ratio, and of course their miniaturised size for interaction with biological cells and analysis of very small sample volumes [1]. Microelectrodes have widely been used in various applications ranging from electrochemical sensing [1–4] to biosensing [5–8] applications. While existing microelectrodes are capable of recording signals at miniaturised size, the development of new microelectrodes is required to further enhance their electrical characteristics (lower electrochemical impedance), particularly for

**Competing interests:** The authors have declared that no competing interests exist.

recording signals from complex neural networks at a single-cell resolution [7, 9]. Hence, research and development of new microelectrodes with different materials and in different topographies have emerged over recent years to lower the impedance of the microelectrodes to improve the signal-to-noise ratio of the electrochemical recordings at miniaturised dimensions [10].

Gold (Au) [11–17], platinum (Pt) [18–20], titanium (Ti) [21, 22] are the most common materials that have been used for microelectrodes due to their excellent electrical characteristics, corrosion resistance and biocompatibility. Platinum black (Pt-black) coatings have also been applied by covering the planar electrodes through a platinisation process to reduce the impedance [12, 23, 24]. Recently, the advent of conductive polymers such as polypyrrole (PPy) [25–27] and poly(3,4ethylenedioxythiophene) (PEDOT) [28–30] as coating layers have also been demonstrated to improve the impedance and biocompatibility of microelectrodes, however, polymer electrodes are prone to mechanically instability, hindering long-term neural applications [31, 32].

As small planar surface areas are required for contemporary biological cell applications, another way to improve the impedance is to increase the 3D surface area by the introduction of 3D topological structures. This has led to the exploitation of 3D nanomaterials to improve the impedance of microelectrodes. Gold nanostructures [13–17], carbon nanotubes (CNTs) [21, 33, 34], vertically aligned silicon nanowires (Si NWs) [35–37], vertically aligned platinum nanowires (Pt NWs) [38, 39] and tin oxide nanowires ($SnO_2$ NWs) [40–43] are the most common nanomaterials that have been used to increase surface area. While these nanomaterials have successfully been used in neural applications, it is difficult to control their fabrication for various topographies, particularly on flexible substrates. In part this is due to the complexity of their fabrication processes such as chemical vapour deposition (CVD) including vapour-liquid-solid growth (VLS) [37] and vapour-solid-solid growth (VSS) [44, 45], electrochemical deposition [14, 15, 17], electroplating [13, 16, 46, 47] or dispersion [34, 48, 49]. All of those methods require fine-tuning, sometimes complex hardware, and often need to be carried out at high temperatures, making integration with pre-existing microelectrode structures challenging.

A more recent introduction to nanowire technology has been the introduction of vertical zinc oxide nanowires (ZnO NWs), an n-type semiconductor, commonly used in light-emitting diodes (LEDs) [50, 51], solar cells [52–56], biosensors [57–59], piezoelectric devices [60–62] and sensing applications [63, 64] due to their excellent electronic and optoelectronic properties. Another favourable property of ZnO NWs is that they can be grown via hydrothermal synthesis at temperatures below 100˚C, which is both a low cost and scalable fabrication method [65]. The hydrothermal synthesis of ZnO NWs allows nanowires to directly grow on flexible substrates, providing an ideal platform for neural applications [57]. Plank *et al.* [66–68] and others [69, 70] have previously demonstrated how the morphology of ZnO NWs can easily be controlled by the hydrothermal growth parameters such as precursor concentration, growth time, growth temperature, ZnO seed layers, and additive auxiliary agents. Furthermore, our recent work has demonstrated how ZnO NWs can be grown on defined regions of a substrate with varied aspect ratios using standard photolithography techniques [66].

ZnO NWs have been demonstrated as a biocompatible material for HEK293 [71], H9c2 cells [72], human NTera2.D1 (hNT) neurons [73] and PC12 neurons [72], however; they have also been shown to reduce the viability of 3T3 fibroblasts [74], human umbilical cord vein endothelial cells [74], bovine capillary endothelial cells [74], hippocampal neurons [75], human hNT neurons [73] primary murine macrophages [76], human monocyte macrophages [77] and osteoblasts [78].

In this article, we show how ZnO NWs can be integrated with microelectrodes. The first encapsulation of ZnO NWs with Cr/Au and PEDOT layers was previously introduced by Ryu *et al.* [71, 79] to improve the electrical characteristics for the recording of neural signals. Expanding on the seminal work of Ryu *et al.* [71, 79], we describe the design and fabrication of metal encapsulated ZnO NW microelectrodes (Cr/Au, Ti and Pt) to improve electrochemical properties of the microelectrodes compared to controls of planar Cr/Au microelectrodes and pristine ZnO NW microelectrodes. In addition, we apply an equivalent circuit model to determine the electrochemical impedance characteristics of the microelectrodes investigated here.

## 2. Experimental details

### 2.1 Fabrication

We fabricated microelectrodes in three different configurations: as a planar Cr/Au microelectrode, as a pristine unfunctionalised vertical ZnO NW microelectrode and as metal encapsulated ZnO NW microelectrodes (Cr/Au-ZnO NW, Ti-ZnO NW and Pt-ZnO NW). The planar microelectrode and pristine ZnO NW microelectrodes were used as controls to investigate the characteristics of the Cr/Au-ZnO NW, Ti-ZnO NW and Pt-ZnO NW microelectrodes. All of the microelectrodes were fabricated on 15 mm × 15 mm silicon substrates with a 100 nm oxide layer (Si/SiO$_2$) (University Wafer) in the same configuration, as shown in Fig 1(A). The layout consisted of 32 central working microelectrodes in a 6 × 6 matrix within an area of 4 × 4 mm$^2$, which were connected to the outer electrode contact pads. The microelectrodes were 200 μm in diameters with spacings of 700 μm in between.

**2.1.1 Planar microelectrode fabrication.** The initial fabrication stages of all microelectrodes involved the production of planar Cr/Au microelectrodes without a SU8 passivation layer, as shown in Fig 1(B). Fig 2 shows the fabrication process of all microelectrodes. The substrates were initially cleaned thoroughly by sonication in acetone (one minute), sonication in isopropanol (IPA) (one minute), rinsing in IPA and then drying in a stream of clean nitrogen (N$_2$). Photolithography was then carried out by coating the substrate with a thin layer of

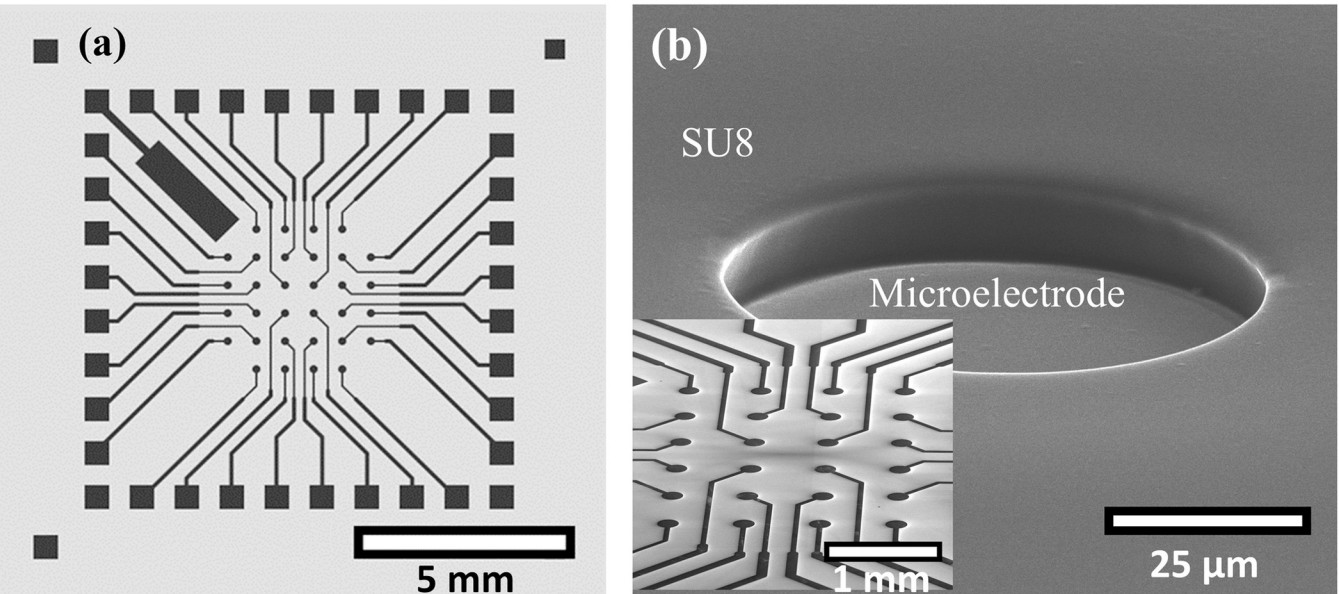

**Fig 1.** (a) Microelectrode configuration. (b) SEM image of fabricated planar microelectrode with a SU8 passivation layer. Inset in (b) shows an SEM image of the planar microelectrode overview without the SU8 passivation layer. The SEM images were taken at 70° tilted-view.

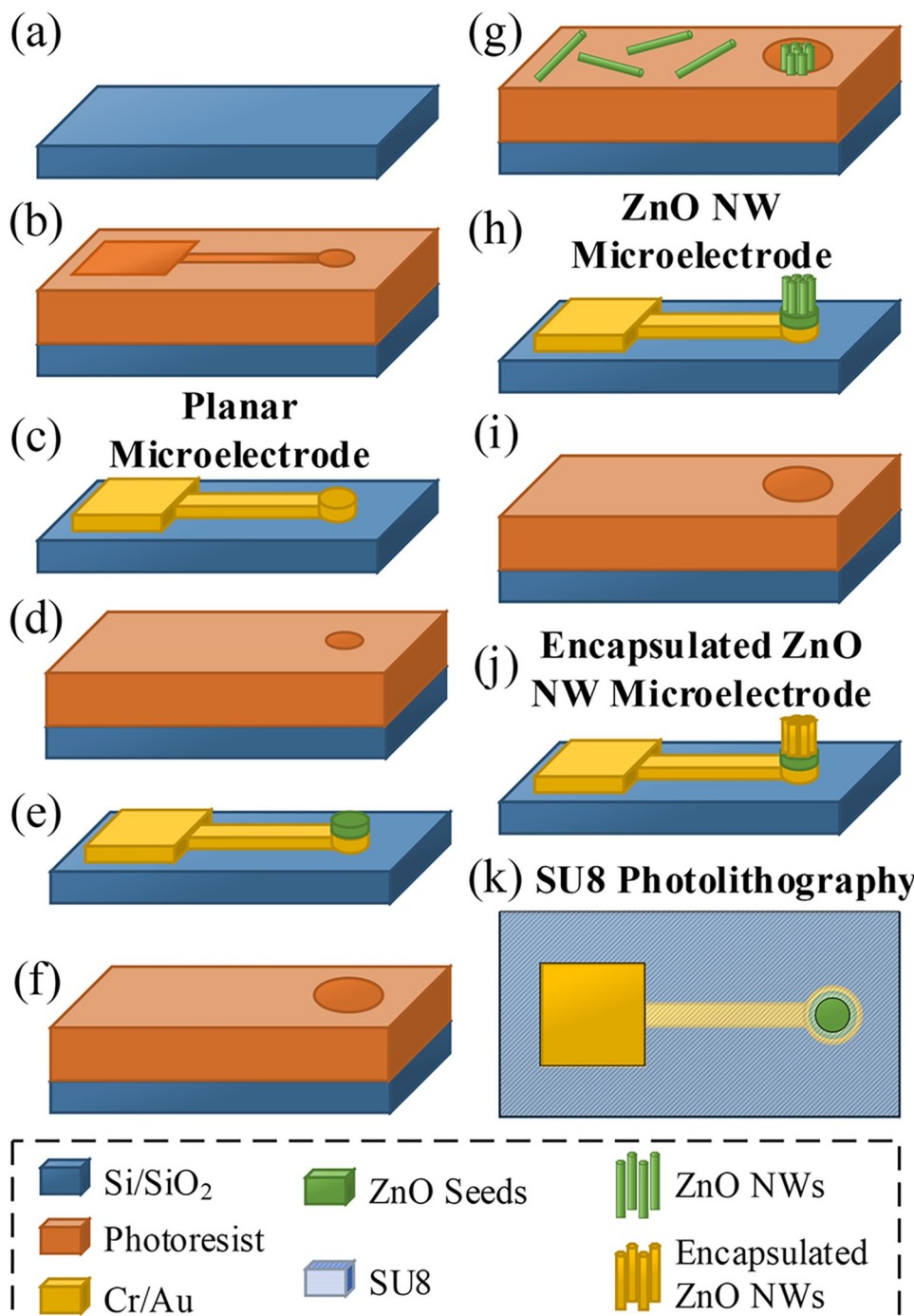

**Fig 2. The fabrication process of the planar microelectrode, pristine ZnO NW microelectrode and metal encapsulated ZnO NW microelectrodes.** (a) Si/SiO$_2$ substrate preparation; (b) photolithography; (c) deposition of Cr/Au layer by thermal evaporation followed by lift-off, resulting in a planar microelectrode; (d) photolithography; (e) Sputter deposition of a ZnO seed layer followed by lift-off; (f) photolithography; (g) hydrothermal growth of ZnO nanowires; (h) lift-off to remove ZnO nanowire debris, resulting in a ZnO NW microelectrode; (i) photolithography; (j) deposition of a metallic encapsulation layer by thermal or e-beam evaporation followed by lift-off, resulting in a metal encapsulated ZnO NW microelectrode. The fabrication of all microelectrodes was completed by SU8 photolithography, as shown from the top-view in (k), which deposited a SU8 passivation layer to cover the entire substrate surface except the regions above the working microelectrodes and outer electrodes for electrical measurements.

AZ1518 photoresist (MicroChemicals) via spin-coating at 4000 rpm for one minute followed by soft-baking at 95˚C for two minutes on a hot-plate. A Karl Suss MJB3 mask aligner is then used to expose defined areas of the photoresist to the UV (250 mJ/cm$^2$). The photoresist development was carried out by agitating in a 1:4 dilution of AZ351B developer (MicroChemicals) to deionised (DI) water for 15–20 seconds followed by 10 seconds of rinsing in DI-water and drying with $N_2$ to define microelectrode patterns, as shown in Fig 2(B). The microelectrodes were deposited via thermal evaporation of 5 nm chrome followed by 50 nm of gold (Cr/Au) onto the Si/SiO$_2$ substrates, using an Angstrom Engineering Nexdep Evaporator. A lift-off process was then applied using acetone and IPA to remove photoresist with the excessive Cr/Au, leaving a pattern of Cr/Au electrodes with central working microelectrodes (200 μm in diameters) on the substrates to create planar microelectrodes, as shown in Fig 2(C).

**2.1.2 ZnO NW microelectrode fabrication.** ZnO NW microelectrodes were fabricated directly onto the planar Cr/Au microelectrodes, as shown schematically in Fig 2(C)–2(H). Photolithography was carried out, again using the AZ1518 photoresist and the Karl Suss MJB3 mask aligner, to define a circular gap of 100 μm diameter above the Cr/Au microelectrodes. A 100 nm thick ZnO seed layer was deposited by sputter deposition, using an HHV Auto 500 RF sputter coater, followed by a lift-off process to leave ZnO seeds on the central working microelectrodes, as shown in Fig 2(E). A third photolithography step was applied using a T-profile AZ5214E (MicroChemicals) recipe by spin-coating at 4000 rpm for one minute, soft-baking at 95˚ for two minutes, UV flash exposure (2.8 mJ/cm$^2$), baking at 110˚ for two minutes, UV mask exposure (224 mJ/cm$^2$), development in 1:4 dilution of AZ351B developer to DI-water for 15–20 seconds followed by 10 seconds rinsing in DI-water and drying with $N_2$ to define a pattern of open areas with diameters of 200 μm concentric with the pre-deposited ZnO seeds, as shown in Fig 2(F).

Hydrothermal synthesis of ZnO NWs, at 95˚C, was then carried out to grow ZnO NWs from the pre-deposited ZnO seed layers. The growth solution was prepared by mixing 25 mM equimolar concentration of zinc nitrate hexahydrate (98%, Sigma Aldrich) and hexamethylenetetramine (HMT 99%, Sigma Aldrich) precursors in 100 mL of DI-water. This concentration was selected to provide the optimum conditions as commonly used in the literature [67, 71, 80, 81] for growing consistent nanowires with relatively high aspect ratios of 25 ± 8 as determined in our previous work [66]. After growing ZnO NWs for 4 h, the samples were cleaned in DI-water and the photoresist was removed by immersion in n-methyl-2-pyrrolidinone (NMP) for 2 h. This was then followed by two minutes of sonication in NMP, two minutes of sonication in IPA, rinsing in IPA and drying with $N_2$. These steps together then resulted in ZnO NW microelectrodes as indicated in Fig 2(H).

**2.1.3 Metal encapsulated ZnO NW microelectrode fabrication.** Metal encapsulated ZnO NW microelectrodes were fabricated by an additional fabrication step on the ZnO NW microelectrodes, as depicted schematically in Fig 2(H)–2(J). Photolithography was again carried out using the AZ1518 photoresist to define open areas with diameters of 200 μm aligned above the ZnO NWs on the central working microelectrodes, as shown in Fig 2(I). Encapsulation layers of either Cr/Au (2/20 nm), Ti (10 nm) or Pt (10 nm) were deposited by thermal or e-beam evaporation, using the Angstrom Engineering Nexdep Evaporator. A lift-off process was then applied to remove the excessive deposited materials, resulting in microelectrodes with metal encapsulated ZnO NWs, as shown in Fig 2(J).

The fabrication of all microelectrodes was completed by deposition of 8 μm thick SU8-2150 (MicroChemicals) as a passivation layer on microelectrodes through a photolithography process, as shown in Fig 2(K). The SU8 layer covered the entire substrate except for the regions above the working microelectrodes (with diameters of 50 μm) and the outer electrodes to allow for electrical measurements. Furthermore, the 8 μm thick SU8 passivation layer assured

that only vertically grown ZnO NWs at the centre of the working microelectrodes were exposed for the electrochemical measurements. The samples were finally hard-baked at 200˚C for 15 minutes to improve the chemical and physical stability of the devices.

## 2.2 Characterisation

Scanning electron microscopy (SEM) was performed using an FEI Nova NanoSEM 450 to characterise microelectrodes and to determine the morphology of the grown ZnO NWs. The diameter, length and density of the ZnO NWs were measured from the SEM images taken both in the top-view and at a 70˚ tilted-view, using the software from ImageJ.

Electrochemical impedance spectroscopy (EIS) measurements of the microelectrodes were taken using an Agilent 4294A precision impedance analyser, as shown in Fig 3. A frame made of polydimethylsiloxane (PDMS, Sylgard 184) was used to keep the phosphate-buffered saline solution (PBS-1x, Sigma Aldrich), typically used for electrochemical measurements and in vitro cell culture, on the central region of the samples excluding the outer electrode contacts. The high potential and current terminals of the impedance analyser were connected to a tungsten electrode probe (Rucker & Kolls) and placed in the PBS solution. The low potential and current terminals were connected to the outer contacts using a tungsten electrode probe. A potential of 100 mV was applied at high terminals by sweeping over a frequency range of 40 Hz to 10 MHz, and the impedance was recorded at the low terminals.

## 2.3 EIS correction

During EIS measurements the observed impedance contains the cumulative impedance of the microelectrode as well as all auxiliary connections used between the impedance analyser and the microelectrode. The auxiliary connections consisted of coaxial cables, connectors, the tungsten probe electrodes and the PBS solution that drew away the measured impedance from the actual impedance of the microelectrode. An open/short compensation was carried out to

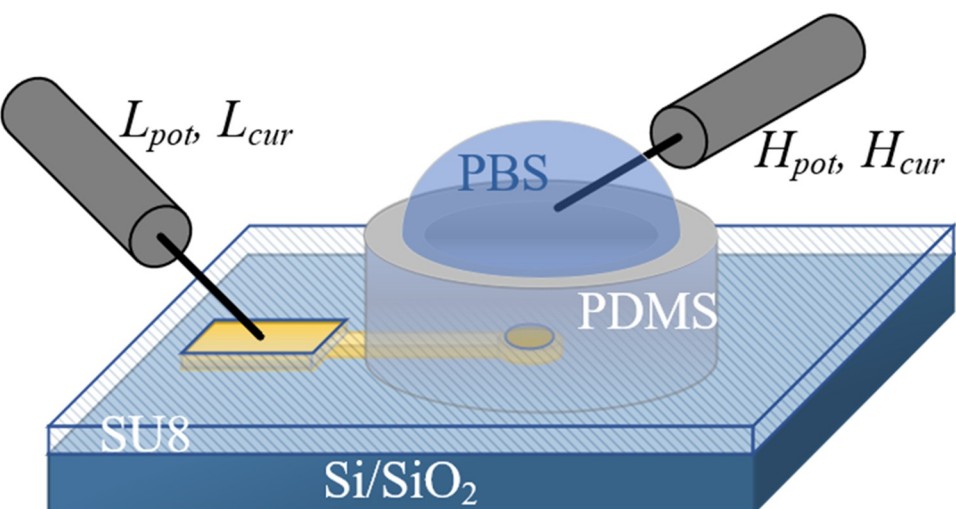

**Fig 3. Electrochemical impedance spectroscopy (EIS) measurement configuration.** Two-point measurement configuration was applied for measuring EIS of each individual microelectrode. Hpot and Hcur represent the high potential and current terminals of the impedance analyser instrument connected to a tungsten electrode probe placed in the phosphate-buffered saline solution (PBS-1x). Lpot and Lcur represent the low potential and current terminals of the instrument connected to another tungsten electrode probe attached to the outer electrode contact. The polydimethylsiloxane (PDMS) frame kept the PBS solution on the central working microelectrodes, excluding the outer electrode contacts.

correct the impedance measurements and to determine the true impedance of the microelectrodes. The corrected impedance was calculated by Eq 1 [82].

$$Z_{ME} = \frac{Z_M - Z_S}{Z_O - Z_M} Z_O,$$ (1)

where $Z_{ME}$ is the true impedance of the microelectrode, $Z_M$ is the measured impedance of the microelectrode, $Z_O$ is the open-circuit impedance and $Z_S$ is the short-circuit impedance. The open-circuit impedance was measured by opening the circuit at the electrode probe ends. The short-circuit impedance was measured by placing both electrode probes in the PBS solution. All of the impedance data presented in this work are the average impedance measurements taken from 20 electrodes ($N \geq 20$) on multiple samples ($M \geq 3$) with the same electrode configuration. The measurements have all been corrected by applying the open/short compensation described above. The impedance data were plotted in the form of Bode plots, showing impedance magnitude vs frequency and impedance phase vs frequency. The frequency of 1 kHz was selected for the comparison of the impedance measurements from different microelectrodes since 1 kHz has commonly been used in neural and sensing applications [9, 30].

## 3. Results and discussion

### 3.1 ZnO NW microelectrodes

Fig 4(A) shows SEM images of ZnO NWs that were grown from a seed layer on the microelectrode. The SEM shows that the ZnO NWs have been grown uniformly with an average diameter of $75 \pm 23$ nm, length of $1.88 \pm 0.17$ μm, and density of $57 \pm 6$ NWs/μm$^2$ within the 100 μm region. While the majority of ZnO NWs were vertically aligned, some NWs were protruding from the outer edges of the seed layer exposed to the growth solution, as shown in Fig 4(A) inset. Fig 4(B) shows the ZnO NW microelectrode after photolithography of the SU8 passivation layer. The SU8 passivation layer covered the entire substrate surface with an open window of 50 μm in diameter above the microelectrode to ensure that only the vertical ZnO NWs within the 50 μm wide area were exposed to the solution for EIS.

Fig 5 shows the SEM image and energy-dispersive X-ray spectroscopy (EDS) of the fabricated ZnO NW microelectrode with the SU8 passivation layer. The EDS mapping shows silicon (Si) of the Si/SiO$_2$ substrate and gold (Au) of the deposited Cr/Au microelectrode in Fig 5 (B) and 5(C), respectively. Since the deposited microelectrode is 200 μm in diameter, gold is appearing everywhere in Fig 5(C). The existence of synthesised ZnO NWs is confirmed by zinc (Zn) and oxygen (O) mapping in Fig 5(D) and 5(E), respectively, however oxygen is also attributed to the Si/SiO$_2$ substrate. Fig 5(F) shows carbon (C) mapping confirming the successful photolithography of the SU8 passivation layer with a 50 μm wide open window above the microelectrode.

Fig 6 shows the EIS Bode plots of planar microelectrodes, microelectrodes with ZnO seed layers and with pristine ZnO NWs that were averaged over multiple samples with the same configuration. The open/short compensation was applied, as previously described in Section 2.3, to determine the true impedance of the microelectrodes, excluding the disturbed impedance from the auxiliary connections, as shown in S1 Fig. Fig 6(A) shows that the impedance magnitude of the microelectrodes decreased with increasing the frequency from 40 Hz to 10 MHz. While adding ZnO seeds to the planar microelectrodes increased the impedance, growing ZnO NWs on microelectrodes reduced the impedance to be lower than the planar microelectrodes, particularly in the low frequency range of 40–2000 Hz, as shown in Fig 6(B).

Fig 6(C) and 6(D) show the average impedance phase of −90˚ for the microelectrodes indicating their dominant capacitive behaviour. The measured impedance phase (without open/

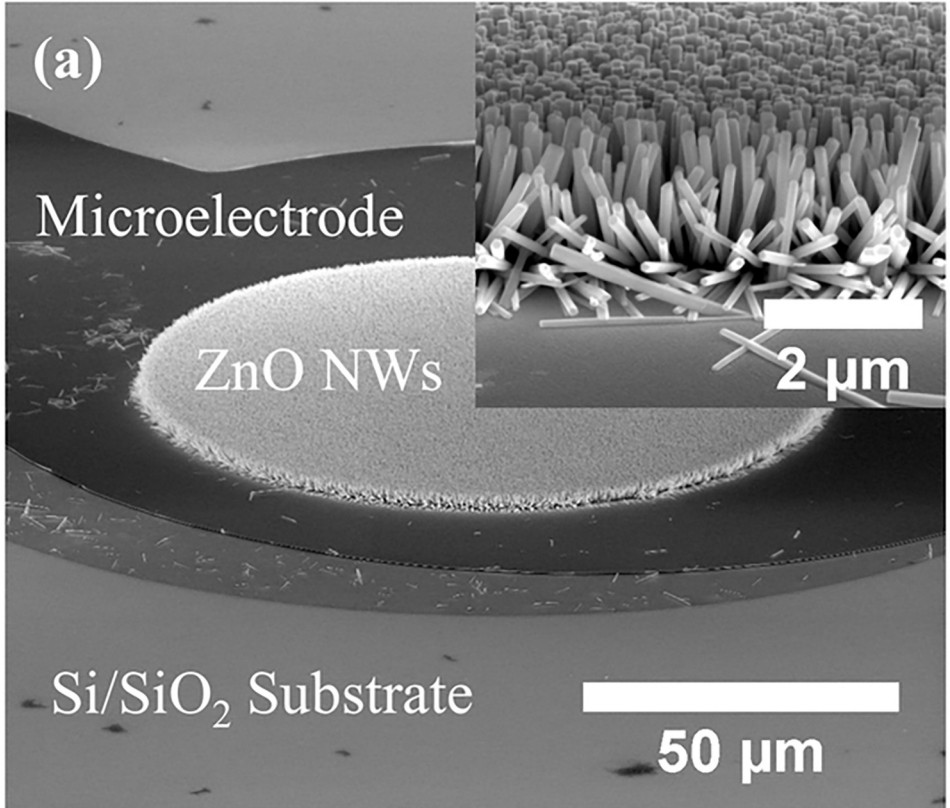

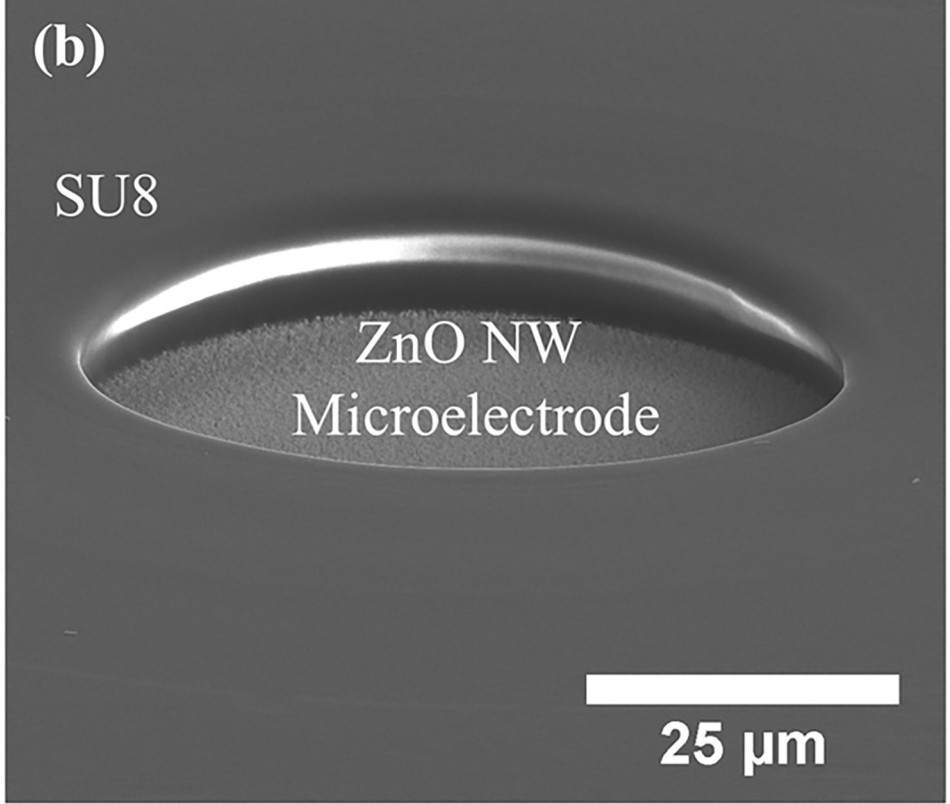

**Fig 4.** (a, b) SEM images of a ZnO NW microelectrode before and after photolithography of a SU8 passivation layer, respectively. Inset in (a) shows an SEM image of the ZnO NWs grown from the seed layer on the microelectrode. The SEM images were taken at 70˚ tilted-view.

short compensation) was demonstrated to be stable at −90˚ for the frequency range of 40 Hz to 100 kHz, as shown in S1(B) Fig. As the frequency increased from 100 kHz to 10 MHz, the measured phase started to approach 0˚ (Ohmic behaviour). However, applying the open/short compensation showed that in fact the impedance phase only reached a maximum of −70˚ at 2 MHz for the planar microelectrodes, as shown in Fig 6(C). The transition of the impedance behaviour from capacitive to ohmic at high frequencies could, therefore, be determined as an influence of the auxiliary connections or the PBS solution, not the characteristics of the microelectrodes.

The frequency of 1 kHz is typically used in neural applications as this is where neural action potentials (APs) typically take place. The electrical characteristics of different microelectrodes can, therefore, be investigated by comparing their impedance at the frequency of 1 kHz. Ganji *et al.* [30] previously determined that the impedance of microelectrodes substantially varied from 1 kΩ to 1 MΩ as the diameter of planar gold electrodes decreased from 2 mm to 20 μm. The fabricated Cr/Au planar microelectrodes in our work were measured to have an average impedance magnitude of 835 ± 40 kΩ with a phase of −89˚ ± 0.5˚ at the frequency of 1 kHz. The measured impedance is higher than the average impedance of 500 kΩ that Ganji *et al.* measured for 50 μm wide gold electrodes, but lower than 1.17 ± 0.24 MΩ that Nick *et al.* [13] measured for 40 μm wide gold electrodes. Furthermore, Ganji *et al.* determined that the electrodes with diameters smaller than 100 μm have capacitive-like behaviour with a phase spectra close to −90˚ for the frequency range of 1 Hz to 10 kHz, in agreement with the impedance phase presented in our work.

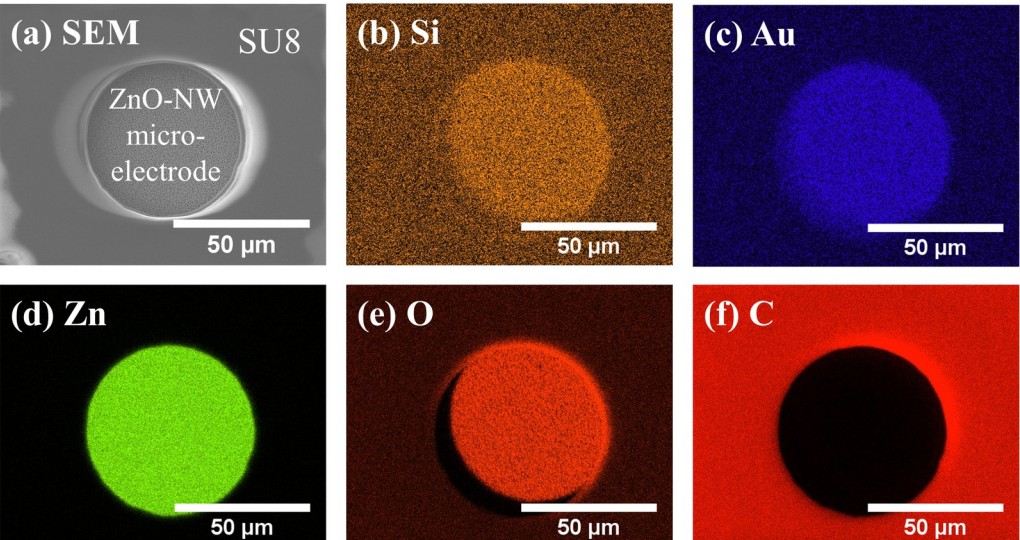

**Fig 5. SEM image and energy-dispersive X-ray spectroscopy (EDS) mapping of a ZnO NW microelectrode.** (a) SEM image of the ZnO NW microelectrode enclosed with a SU8 passivation layer. (b)–(f) EDS mapping of the ZnO NW microelectrode shown in (a). The EDS mapping shows (b) silicon (Si) of the Si/SiO$_2$ substrate; (c) gold (Au) of the underlying Cr/Au microelectrode; (d) zinc (Zn) of ZnO nanowires; (e) oxygen (O) of ZnO nanowires and the Si/SiO$_2$ substrate; and (f) carbon (C) of the SU8 passivation layer.

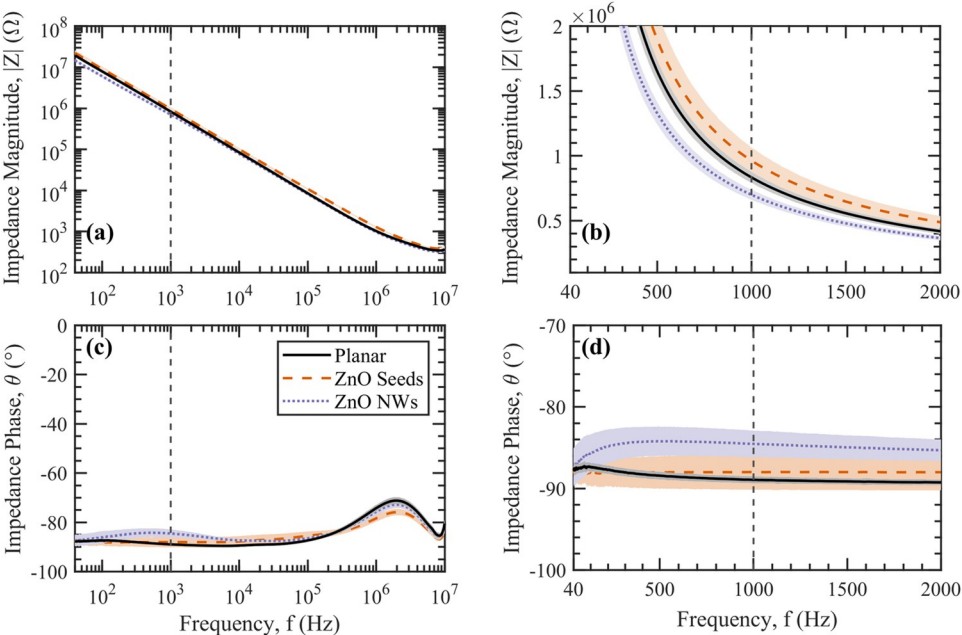

**Fig 6. Electrochemical impedance spectroscopy (EIS) of planar microelectrodes, microelectrodes with ZnO seed layers and with pristine ZnO NWs.** (a, b) Impedance magnitude and (c, d) impedance phase are plotted vs frequency with dashed lines indicating the frequency of 1 kHz. (b) and (d) show the electrochemical impedance magnitude and phase, respectively, for the frequency range of 40 Hz to 2 kHz in linear scale. Error shades represent one standard deviation (N ≥ 20 electrodes).

The deposition of a 100 nm thick sputtered ZnO seed layer onto the planar microelectrodes increased the average impedance to 965 ± 100 kΩ with a phase of −88° ± 2° at 1 kHz. We presume that the semiconductor-like behaviour of the polycrystalline ZnO seed layer has resulted in the increased impedance of the microelectrodes in comparison to the metallic planar microelectrodes. In contrast, as single-crystalline ZnO NWs were grown from the seed layer [68, 83–85], the impedance reduced to 700 ± 40 kΩ with a phase of −85° ± 1.5° at 1 kHz. Burke-Govey *et al.* [68] previously demonstrated the high conductivity of the ZnO NWs as the channel in field-effect transistors, where the conductivity of the as-grown ZnO NWs was demonstrated to be less dependent on the field compared to the similarly prepared thinner nanowires. The reduced impedance of the ZnO NWs also agrees with the fact that ZnO NWs have been previously used as collector electrodes in dye-sensitised solar cells (DSSC) [57, 86, 87].

The ZnO NWs that were grown in our work resulted in impedance values that were significantly higher than that observed by Ryu et al. [71, 79]. The current work produced impedance values of 700 ± 40 kΩ at 1 kHz compared to the average impedance of 1.45 kΩ by Ryu et al.. However, the electrodes that they used were significantly larger (800 × 800 μm) than the microelectrodes used in our work (50 μm in diameter). Understanding that the electrochemical impedance is inversely proportional to the electrode base area [30], the multiplication of the impedance magnitude and the electrode base area as an area impedance factor can be used for comparing the electrical characteristics of microelectrodes in the literature. Consequently, the ZnO NWs that were grown on the microelectrodes in our work resulted in an area impedance of 13.7 ± 0.79 Ω·cm² , which is comparable to 9.28 Ω·cm² that Ryu *et al.* reported. Furthermore, the ZnO NWs on our microelectrodes have an average length of 1.88 ± 0.178 μm, which resulted in a relatively smaller surface area and therefore higher area impedance compared to the 3.5 μm long ZnO NWs that Ryu *et al.* used.

## 3.2 Metal encapsulated ZnO NW microelectrodes

Metal encapsulated ZnO NW microelectrodes were fabricated by deposition of Cr/Au (2/20 nm), Ti (10 nm) or Pt (10 nm) onto the ZnO NWs as described in Section 2.1.3. Fig 7(A)–7(D) show top-view SEM images of pristine ZnO NWs, Cr/Au-ZnO NWs, Pt-ZnO NWs and Ti-ZnO NWs, respectively. The metallic layer was deposited perpendicular to the substrate plane, resulting in the tops of the NWs being fully encapsulated by the metallic layer, however, the lower regions are only partially covered, as shown in S2 Fig. Since the metallic encapsulation layer did not cover the entire NW surfaces, the electrochemical conduction path is considered to pass through the ZnO NWs from the solution to the underlying electrodes. The metal encapsulated NWs are therefore thought to behave differently from the pristine metallic NWs (i.e. Cr/Au NWs, Pt NWs and Ti NWs).

Fig 8 shows the EIS Bode plots of microelectrodes with ZnO NWs that were encapsulated with layers of different metals: Cr/Au, Ti, and Pt. Fig 8(A) and 8(B) show that the impedance magnitudes of the Ti and Pt encapsulated ZnO NW microelectrodes were lower, particularly at low frequencies of 40 Hz–2 kHz, compared to the control planar microelectrodes and the ZnO NW microelectrodes with no metal coating. Applying ZnO NWs encapsulated with a 10 nm thin layer of Ti or Pt onto the microelectrodes improved the impedance by a factor of 2×, from 835 ± 40 kΩ of planar microelectrodes to 400 ± 25 kΩ at 1 kHz of frequency. While Ti and Pt coating resulted in improved impedance, the impedance of Cr/Au encapsulated ZnO NW microelectrodes, with an average impedance magnitude of 680 ± 10 kΩ and phase of −84˚

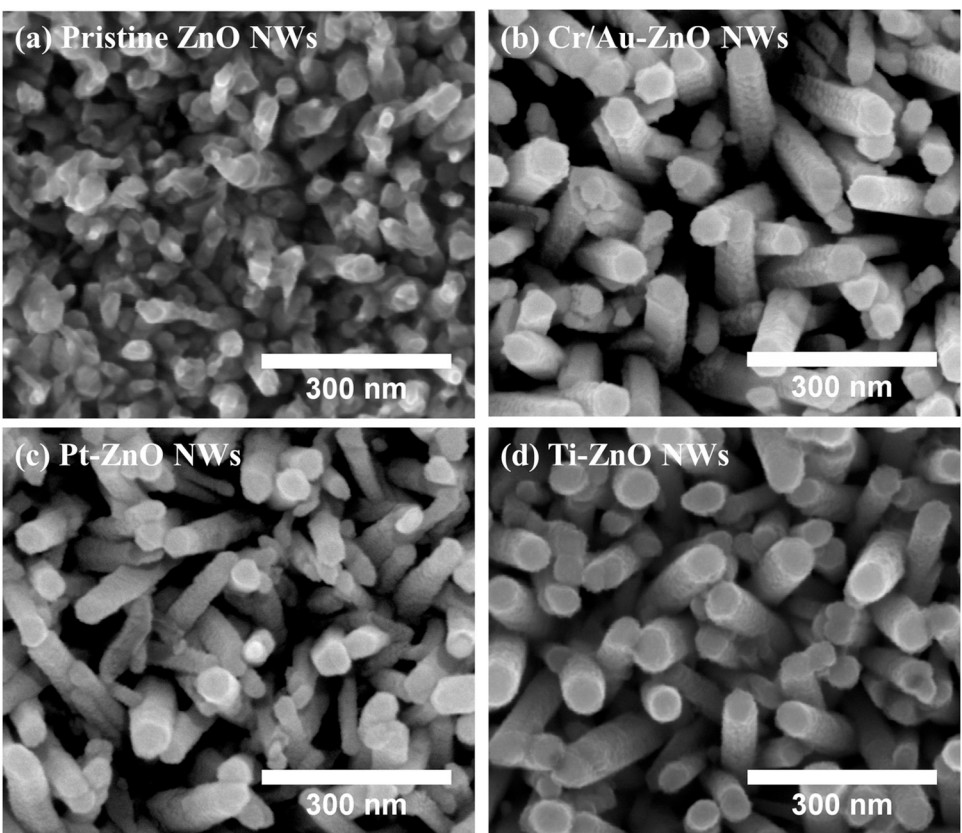

**Fig 7.** SEM images of (a) pristine ZnO NWs, (b) Cr/Au-ZnO NWs, (c) Pt-ZnO NWs, and (d) Ti-ZnO NWs. The SEM images were taken from the top-view.

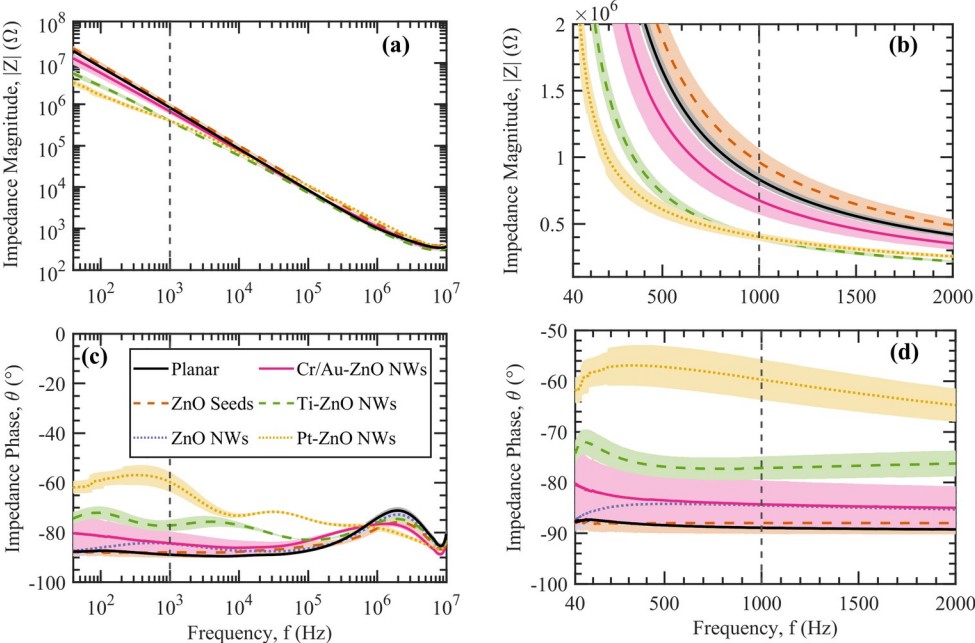

**Fig 8. Electrochemical impedance spectroscopy (EIS) of microelectrodes with metal encapsulated ZnO NWs.** (a, b) Impedance magnitude and (c, d) impedance phase are plotted vs frequency with dashed lines indicating the frequency of 1 kHz. (b) and (d) show the electrochemical impedance magnitude and phase, respectively for the frequency range of 40 Hz to 2 kHz in linear scale. Error shades represent one standard deviation (N ≥ 20 electrodes).

± 5˚, was very similar to the pristine ZnO NW microelectrodes, with an average impedance magnitude of 700 ± 40 kΩ and phase of −85˚ ± 2˚ at 1 kHz. Furthermore, the impedance phase of the Ti and Pt encapsulated ZnO NW microelectrodes were determined to be higher for the frequencies below 100 kHz compared to the other microelectrodes, as shown in Fig 8(C). The impedance phase of Ti and Pt encapsulated ZnO NW microelectrodes were measured to be −77˚ ± 2˚ and −60˚ ± 4˚ at 1 kHz, respectively, as shown in Fig 8(D).

Pristine Au and Pt NWs were previously reported to have an electrochemical impedance within the range of 13–500 kΩ (0.092–9.82 Ω·cm$^2$) [13, 17] and 2 kΩ–6 MΩ (0.226–6 Ω·cm$^2$) [38, 39], respectively, for different morphologies of the nanowires. The ZnO NWs that were encapsulated with Cr/Au and Pt in our work resulted in a slightly higher area impedance of 13.4 ± 0.2 Ω·cm$^2$ and 7.85 ± 0.49 Ω·cm$^2$, respectively than the pristine metallic NWs. This is therefore due to the incorporation of the semiconducting ZnO NWs that resulted in a relatively higher impedance than the pristine metallic NWs.

### 3.3 EIS modelling

A modified Randles equivalent circuit model [30, 88–90], as shown in Fig 9, was constructed to theoretically compare the EIS of the control planar and ZnO NW microelectrodes with the metal encapsulated ZnO NW microelectrodes. The equivalent circuit comprises a series resistor (R$_s$), charge transfer resistor (R$_{ct}$), adsorption capacitor (C$_{ad}$), and constant phase element (CPE). The series resistor (R$_s$) represents all series resistance in the EIS measurement, including the resistance of the auxiliary connections, PBS solution, and underlying metallic contacts. The charge transfer resistance or the redox reaction resistance (R$_{ct}$) represents the resistance for the charges transferring from the solution to the electrode. The adsorption capacitor (C$_{ad}$) accounts for the adsorption of the transferred species through the double-layer interface onto

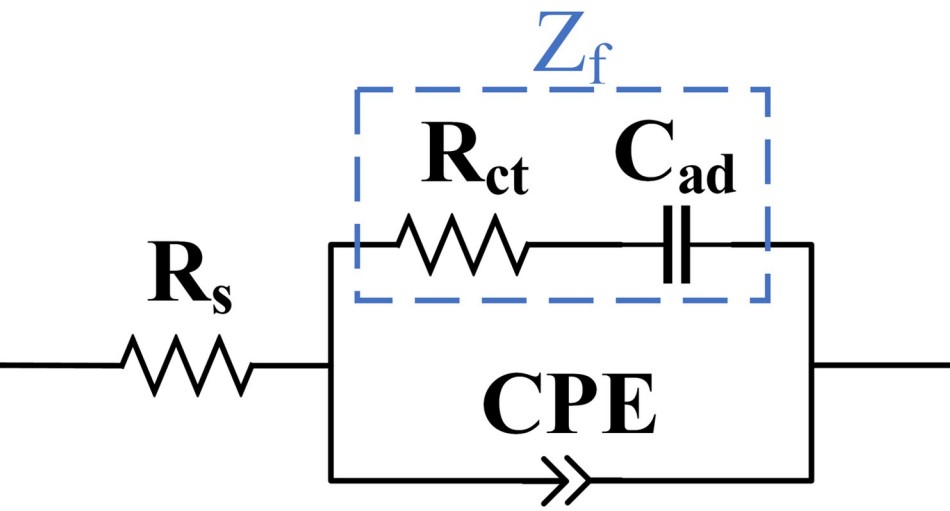

**Fig 9. Modified Randles equivalent circuit model of microelectrodes for electrochemical impedance modelling.**
The equivalent circuit comprises a series resistor ($R_s$), charge transfer resistor ($R_{ct}$), adsorption capacitor ($C_{ad}$), and constant phase element (CPE). $Z_f$ represents the Faradaic impedance of the circuit as a series combination of $R_{ct}$ and $C_{ad}$ impedance.

the electrode surface that participates in the redox reaction. The Faradaic impedance, $Z_f$, that is representative of all redox reactions at the electrode-electrolyte interface can subsequently be defined as a series combination of the charge transfer resistance and the adsorption capacitance by

$$Z_f = R_{ct} + jX_{Cad},\qquad(2)$$

where $X_{Cad}$ is the reactance associated with the adsorption capacitance.

The constant phase element (CPE) accounts for the double-layer capacitance at the interface of the electrode and the solution that behaves as a non-Faradaic pseudo-capacitance. The reactance of the CPE is defined as

$$|X_{CPE}| = \frac{1}{Q\,(2\pi f)^n},\qquad(3)$$

where n is the constant exponent, $0 \leq n \leq 1$, with 1 representing an ideal capacitor and 0 representing an ideal resistor. The Q is the constant coefficient with the unit of $F \cdot cm^{-2} \cdot s^{n-1}$ that represents the double-layer capacitance at the angular frequency of 1 rad/s.

The obtained impedance of microelectrodes was fitted to the equivalent circuit by using the Palmsens PSTrace software to extract the electrochemical parameters of the microelectrode impedance, as detailed in S1 File. Table 1 shows the impedance parameters of the microelectrodes that were determined by fitting to the modified Randles equivalent circuit. A Chi-Squared test was applied to evaluate the goodness of the fit, ensuring that the average error of the fit is less than 1%. The comparison between the impedance of the simulated circuit ($Z_{sim}$), as detailed in the Supporting Information, and the measured impedance ($Z_{ME}$) also confirms the success of the EIS modelling using the modified Randles circuit. The series resistance ($R_s$) was determined to be negligible (1 n$\Omega$) for all microelectrodes. The small series resistance demonstrates the successful impedance correction achieved by the open/short circuit compensation, diminishing the influences of the solution and the auxiliary connections on the impedance measurements.

**Table 1. Impedance parameters of the microelectrodes.** Parameters are measured by fitting the measured impedance (after the open/short circuit compensation) to the modified Randles equivalent circuit.

| Microelectrode | $\lvert Z_{ME}\rvert$[a] | $\theta_{ME}$[a] | $R_{ct}$[b] | $C_{ad}$[b] | $\lvert Z_f\rvert$[b] | $\lvert X_{CPE}\rvert$[b] | $\lvert Z_{sim}\rvert$[b] |
|---|---|---|---|---|---|---|---|
| Planar | 835 kΩ | −89° | 1.27 kΩ | 97.1 pF | 1.64 MΩ | 1.66 MΩ | 825 kΩ |
| ZnO seeds | 965 kΩ | −88° | 2.01 kΩ | 58.1 pF | 2.74 MΩ | 1.50 MΩ | 969 kΩ |
| ZnO NWs | 700 kΩ | −85° | 1.46 kΩ | 77.5 pF | 2.05 MΩ | 1.03 MΩ | 686 kΩ |
| Cr/Au-ZnO NWs | 680 kΩ | −84° | 3.04 kΩ | 57.7 pF | 2.76 MΩ | 871 kΩ | 662 kΩ |
| Ti-ZnO NWs | 400 kΩ | −77° | 2.27 kΩ | 36.6 pF | 4.35 MΩ | 432 kΩ | 393 kΩ |
| Pt-ZnO NWs | 405 kΩ | −60° | 1.85 MΩ | 66.8 pF | 2.09 MΩ | 474 kΩ | 420 kΩ |

[a] Measured impedance parameters at the frequency of 1 kHz.

[b] Simulated equivalent circuit parameters at the frequency of 1 kHz.

The planar microelectrodes were determined to have an equal contribution of the Faradaic and non-Faradic electrochemical impedance (ca. 1.65 MΩ at 1 kHz), through $\lvert Z_f\rvert$ and $\lvert X_{CPE}\rvert$, respectively. In the case of a ZnO seed layer deposited on the planar microelectrode, the Faradaic impedance increased to 2.74 MΩ at 1 kHz due to its semiconductive characteristics. The overall impedance of the microelectrodes with ZnO seeds was subsequently higher than the planar microelectrodes.

The growth of ZnO NWs from the seed layer has reduced the 1 kHz impedance from 965 ± 100 kΩ of ZnO seeds to 700 ± 40 kΩ. We propose that this is primarily due to the increased surface area and, secondarily, to direct conduction pathways. The resulting ZnO NWs were measured from SEM images to have an average diameter of 75 ± 23 nm, length of 1.88 ± 0.17 μm, and density of 57 ± 6 NWs/μm². The average surface area of the microelectrode was subsequently increased by a factor of 25×, from 1,960 μm² of the planar microelectrode to 50,400 μm² by growing ZnO NWs. The average surface area was estimated assuming ZnO NWs as cylinders, as described in S1 File. Such an increase of the microelectrode surface area has increased the adsorption capacitance ($C_{ad}$) that is associated with the redox reaction and the double-layer capacitance (CPE). Subsequently, the $C_{ad}$ reactance measurement has decreased from 2.74 MΩ of the microelectrode with ZnO seed layer to 2.05 MΩ of ZnO NWs and the CPE reactance from 1.50 MΩ to 1.03 MΩ, respectively. Furthermore, the growth of ZnO NWs is thought to increase the electron transport by providing the direct conduction pathways from the nanowire surface to the underlying microelectrodes [87, 90–94]. As a result, the charge transfer resistance ($R_{ct}$) has been reduced from 2.01 kΩ of ZnO seeds to 1.46 kΩ when ZnO NWs were grown on microelectrodes.

The encapsulation of the ZnO NWs with Pt, Cr/Au and Ti layers increased the 1 kHz Faradaic impedance to 2.09 MΩ, 2.76 MΩ and 4.35 MΩ, respectively. The redox reaction taken at the surface of the metal encapsulated ZnO NWs was reduced as the charge transfer resistance increased and the adsorption capacitance decreased. The charge transfer resistance was significantly increased from 1.46 kΩ to 1.85 MΩ when Pt encapsulation was applied. However, since the Faradaic impedance was dominantly controlled by the adsorption capacitance at low frequencies, ≤ 1kHz, the overall Faradaic impedance did not vary substantially.

The non-Faradaic impedance of the metal encapsulated ZnO NWs was determined to be the dominant factor that improved the EIS of the microelectrodes. The application of Ti and Pt encapsulation on the ZnO NWs reduced the 1 kHz non-Faradaic impedance from 1.03 MΩ of the pristine ZnO NWs to 432 kΩ and 474 kΩ, respectively, which improved their overall electrochemical impedance. The SEM and TEM micrographs in S3(B) and S3(C) Fig show that applying a 10 nm thin encapsulation layer of Pt and Ti created a non-uniform layer on the

NW surfaces that further increased the overall surface area and subsequently reduced the non-Faradaic impedance. The Cr/Au encapsulation of ZnO NWs also reduced the non-Faradaic impedance to 871 kΩ, but with less impact compared to the Ti and Pt encapsulation. As a thick layer was deposited for the Cr/Au encapsulation (2/20 nm), the encapsulation layer could cover the NWs more uniformly (lower porosity), as shown by SEM in S3(A) Fig. Applying the thick encapsulation layer of Cr/Au on the NWs could also fuse the adjacent NWs, particularly at the top regions that reduced the overall surface area of the Cr/Au encapsulated NWs compared to the Ti and Pt encapsulated NWs.

The integration of ZnO NWs with microelectrodes is demonstrated to reduce the impedance compared to the planar microelectrodes due to the increased 3D surface area. Expanding the seminal work of Ryu *et al.* [71, 79], we show that the encapsulation of ZnO NWs with different metals (Ti, Pt) also improve the sensitivity of the microelectrodes due to the reduced electrochemical impedance. Furthermore, the miniaturised size of the metal encapsulated microelectrodes can provide alternative microelectrodes for biological applications to record signals of single-cells at high signal-to-noise ratio.

## 4. Conclusion

In this article, we demonstrate how the encapsulation of the ZnO NWs with different metals (Cr/Au, Ti, Pt) demonstrates a further decrease in the electrochemical impedance of the microelectrodes. The Ti and Pt encapsulated NWs notably improved the impedance of the microelectrodes by a factor of 2× at low frequencies, i.e. 400 ± 25 kΩ at 1 kHz of frequency where neural activity is typically recorded. Applying a modified Randles equivalent circuit model demonstrated that the improved impedance of the metal encapsulated ZnO NWs is due to the reduced non-Faradaic impedance, i.e. increased surface area, at the electrode-electrolyte interface. In addition to the ability to tune the growth of the ZnO NWs accurately, the results reported here demonstrated that the Ti and Pt encapsulated ZnO NWs offer an alternative viable microelectrode modality that is attractive for *in vitro* biological cell applications, permitting single-cell resolution recording and high signal-to-noise ratio.

## Supporting information

**S1 Fig.** Electrochemical impedance spectroscopy bode plots as (a) impedance magnitude vs frequency and (b) impedance phase vs frequency of planar microelectrodes. The average impedance of planar microelectrodes before and after the open/short compensation are indicated by 'MEA-No comp' and 'MEA-O/S comp', respectively. The measured impedance of the open-circuit 'O-Circuit' and the short-circuit 'S-Circuit' that were used for the open/short compensation are also plotted in (a) and (b). Dashed lines indicate the frequency of 1 kHz.
(TIF)

**S2 Fig.** Cross-sectional scanning electron microscopy (SEM) of ZnO NWs (a) before and (b) after encapsulation with Cr/Au (2/20 nm) through the thermal evaporation that was applied perpendicular to the substrate plane. While the tops of the NWs are fully encapsulated, the lower regions are partially covered by the Cr/Au layer.
(TIF)

**S3 Fig.** Scanning electron microscopy (SEM) of ZnO NWs encapsulated with (a) Cr/Au (2/20 nm) and (b) Pt (10 nm). (c) Transmission electron microscopy (TEM) of ZnO NWs encapsulated with Ti (10 nm).
(TIF)

**S1 Table. Literature comparison of nanomaterial microelectrodes.**
(PDF)

**S2 Table. Impedance parameters of the microelectrodes.** Parameters are measured by fitting the measured impedance (after the open/short circuit compensation) to the modified Randles equivalent circuit.
(TIF)

**S1 File. Equivalent circuit model derivations.**
(PDF)

## Acknowledgments

We would greatly acknowledge the support from Victoria University of Wellington, the University of Auckland and the MacDiarmid Institute of New Zealand.

## Author Contributions

**Conceptualization:** Mohsen Maddah, Charles P. Unsworth, Gideon J. Gouws, Natalie O. V. Plank.

**Data curation:** Mohsen Maddah.

**Formal analysis:** Mohsen Maddah.

**Funding acquisition:** Charles P. Unsworth, Natalie O. V. Plank.

**Investigation:** Mohsen Maddah.

**Methodology:** Mohsen Maddah, Gideon J. Gouws.

**Project administration:** Charles P. Unsworth, Natalie O. V. Plank.

**Resources:** Gideon J. Gouws.

**Supervision:** Charles P. Unsworth, Natalie O. V. Plank.

**Writing – original draft:** Mohsen Maddah.

**Writing – review & editing:** Mohsen Maddah, Charles P. Unsworth, Gideon J. Gouws, Natalie O. V. Plank.

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
