## [Decision Letter · Decision Letter 0]

31 Mar 2022

PONE-D-22-07435Synthesis of encapsulated ZnO nanowires provide low impedance alternatives for microelectrodes 

Dear Dr. Maddah,

Thank you for submitting your manuscript to PLOS ONE. After careful consideration, we feel that it has merit but does not fully meet PLOS ONE’s publication criteria as it currently stands. Therefore, we invite you to submit a revised version of the manuscript that addresses the points raised during the review process.

ACADEMIC EDITOR:The comments received from reviewers are given below:

We look forward to receiving your revised manuscript.

Kind regards,

Dr. Sandeep Arya

Academic Editor

PLOS ONE

Journal Requirements:

"This research was supported by the Royal Society of New Zealand Marsden Fund (3709273/UOA1510). We would also greatly acknowledge the support from Victoria University of Wellington, the University of Auckland and the MacDiarmid Institute of New Zealand."

"Mohsen Maddah: This research was supported by the Royal Society of New Zealand Marsden Fund (3709273/UOA1510)."

Additional Editor Comments:

Comments from Reviewer 2:

Authors carried out their work on -Synthesis of encapsulated ZnO nanowires provide low impedance alternatives for microelectrodes. The manuscript is technically good but there are certain queries that need to be addressed as follows:

1. Authors should give a detailed description of the fabrication step of the microelectrode as the complete manuscript revolves around it.

2. Experiment conditions should be optimized, including the pH values and concentration of precursors.

3. Figure 1 is very poorly described. This figure should be better described.

4. SEM analysis needs more elaboration. Why were some nanowires protruding from the outer edges of the seed layer?

5. Why the TEM and SEAD analyses were not done? Why the point-EDS analyses were not done?

6. XRD should also be included in the manuscript.

7. Authors should add high quality figures in the manuscript.

8. Authors should explain the reasons that lead to lowest electrochemical impedance on encapsulation of ZnO NW microelectrodes with thin layer of Ti or Pt?

9. Authors should also encapsulate ZnO NW microelectrode with thin layer of Au, as their control study.

10. Some related papers based on nanowires and electrochemical sensors should be cited to give a broader view of the corresponding research field such as https://doi.org/10.1016/j.microc.2020.104858,

https://doi.org/10.1149/1945-7111/abdee8, https://doi.org/10.1016/j.jmrt.2020.10.024,

https://doi.org/10.1007/s00339-018-1968-8.

Reviewers' comments:

Reviewer's Responses to Questions

**Comments to the Author**

1. Is the manuscript technically sound, and do the data support the conclusions?

Reviewer #1: Yes

Reviewer #2: Yes

Reviewer #3: Yes

2. Has the statistical analysis been performed appropriately and rigorously? 

Reviewer #1: Yes

Reviewer #2: Yes

Reviewer #3: Yes

3. Have the authors made all data underlying the findings in their manuscript fully available?

Reviewer #1: No

Reviewer #2: Yes

Reviewer #3: Yes

4. Is the manuscript presented in an intelligible fashion and written in standard English?

Reviewer #1: Yes

Reviewer #2: Yes

Reviewer #3: Yes

6. PLOS authors have the option to publish the peer review history of their article (what does this mean?). If published, this will include your full peer review and any attached files.

Reviewer #1: No

Reviewer #2: No

Reviewer #3: No

5. Review Comments to the Author

Reviewer #1: In this research article author has synthesized encapsulated ZnO nanowires which is used as a low impedance microelectrode. However, there are various shortcoming in the manuscript which are mentioned below. After major revision this manuscript is suitable for publishing.

1. XRD of synthesized ZnO nanowires is not present in the manuscript. Without XRD how author claim that the synthesized material is ZnO. XRD should be required in the revised manuscript.

2. All the Figures have very low Quality. Enhancement in the quality of images is required in the manuscript.

3. The comparison table of the synthesized materials with the previously reported literature must be required in the revised version.

4. How the improvement in redox reaction of the NWs affect only the Faradaic impedance.

5. How ZnO nanowires provide low impedance, authors are required to describe its mechanism.

6. Authors had tried to explain the results of Figure 6, but the explanation is not sufficient related with these results.

7. In the section of “Modified Randles equivalent circuit model of microelectrodes for electrochemical impedance modelling” author have not given any reference. And the explanation of these equivalent circuit needs some support from the previously reported work.

8. Add some latest reference in the manuscript like ACS Applied Electronic Materials 2, 3522−3529: ECS Journal of Solid State Science and Technology 10, 023002: IEEE Journal of Photovoltaics 10 (6), 1744-1749: Journal of Alloys and Compounds 814, 152292: Optical Materials 79, 115-119

Reviewer #3: (Research Article): PLOS ONE, PONE-D-22-07435

Synthesis of encapsulated ZnO nanowires provide low impedance alternatives for microelectrodes

Comments:

The author presented the synthesis of Ti and Pt encapsulated ZnO nanowires (Ti-ZnO NWs and Pt-ZnO NWs). And their application as Microelectrodes has been presented.

The manuscript is fascinating and well prepared.

My comments are not critical but could help to improve the manuscript.

1. In abstract, author mentioned the “seminal work of Ryu et al. by demonstrating h….” (Page 7, line 20). I think it is good to not to mention the reference. It could cover in the last of introduction part.

2. It is a bit confusing about the control sample. Why there are three control samples, planar Cr/Au, pristine ZnO NWs and Cr/Au-ZnO NWs)?

3. The synthesis part is a bit confusing. It could be good to prepare a separate section in experimental details part to cover the synthesis part.

4. It is good to replace the “Our group” by mentioned the author name et al. Please check, Line number 235, Page 16.

5. It is good to put the Ryu et al. and author comparative values into a Table.

6. A few Figures are not clear. Please check their quality.

---

## [Author Response · Author response to Decision Letter 0]

22 May 2022

We would like to thank you all reviewers for reviewing our manuscript and giving constructive feedbacks. We have amended the entire manuscript as requested.

---

## [Decision Letter · Decision Letter 1]

6 Jun 2022

Synthesis of encapsulated ZnO nanowires provide low impedance alternatives for microelectrodes

PONE-D-22-07435R1

Dear Dr. Mohsen Maddah,

We’re pleased to inform you that your manuscript has been judged scientifically suitable for publication and will be formally accepted for publication once it meets all outstanding technical requirements.

Kind regards,

Dr. Sandeep Arya

Academic Editor

PLOS ONE

Additional Editor Comments (optional):

NA

Reviewers' comments:

Reviewer's Responses to Questions

**Comments to the Author**

1. If the authors have adequately addressed your comments raised in a previous round of review and you feel that this manuscript is now acceptable for publication, you may indicate that here to bypass the “Comments to the Author” section, enter your conflict of interest statement in the “Confidential to Editor” section, and submit your "Accept" recommendation.

Reviewer #1: All comments have been addressed

Reviewer #3: All comments have been addressed

2. Is the manuscript technically sound, and do the data support the conclusions?

Reviewer #1: Yes

Reviewer #3: Yes

3. Has the statistical analysis been performed appropriately and rigorously? 

Reviewer #1: Yes

Reviewer #3: Yes

4. Have the authors made all data underlying the findings in their manuscript fully available?

Reviewer #1: Yes

Reviewer #3: Yes

5. Is the manuscript presented in an intelligible fashion and written in standard English?

Reviewer #1: Yes

Reviewer #3: Yes

6. Review Comments to the Author

Reviewer #1: Author have revised the manuscript with greater precision, Now the manuscript is suitable for publication.

Reviewer #3: (No Response)

7. PLOS authors have the option to publish the peer review history of their article (what does this mean?). If published, this will include your full peer review and any attached files.

Reviewer #1: No

Reviewer #3: No

---

## [Editor Report · Acceptance letter]

8 Jun 2022

PONE-D-22-07435R1 

Synthesis of encapsulated ZnO nanowires provide low impedance alternatives for microelectrodes 

Dear Dr. Maddah:

I'm pleased to inform you that your manuscript has been deemed suitable for publication in PLOS ONE. Congratulations! Your manuscript is now with our production department. 

Kind regards, 

on behalf of

Dr. Sandeep Arya 

Academic Editor

PLOS ONE